# PRESTO: A Multilingual Dataset for Parsing Realistic Task-Oriented Dialogs

**Rahul Goel**[*], **Waleed Ammar**[*†♡], **Aditya Gupta**[*], **Siddharth Vashishtha**[*‡♣],
**Motoki Sano**[*], **Faiz Surani**[*‡◇], **Max Chang**, **HyunJeong Choe**, **David Greene**, **Kyle He**,
**Rattima Nitisaroj**, **Anna Trukhina**, **Shachi Paul**, **Pararth Shah**, **Rushin Shah**, **Zhou Yu**[♠]
Google Inc. (`presto.dataset@google.com`)
[♣]University of Rochester  [◇]University of California, Santa Barbara
[♠]Columbia University  [♡]Holistic Intelligence for Global Good

## Abstract

Research interest in task-oriented dialogs has increased as systems such as Google Assistant, Alexa and Siri have become ubiquitous in everyday life. However, the impact of academic research in this area has been limited by the lack of datasets that realistically capture the wide array of user pain points. To enable research on some of the more challenging aspects of parsing realistic conversations, we introduce PRESTO[1], a public dataset of over 550K contextual multilingual conversations between humans and virtual assistants. PRESTO contains a diverse array of challenges that occur in real-world NLU tasks such as disfluencies, code-switching, and revisions. It is the only large-scale human generated conversational parsing dataset that provides structured context such as a user's contacts and lists for each example. Our mT5 model-based baselines demonstrate that the conversational phenomena present in PRESTO are challenging to model, which is further pronounced in a low-resource setup.

## 1 Introduction

Virtual dialog agents (a.k.a. "*assistants*") are increasingly becoming a part of our everyday life. From setting up alarms to controlling one's home environment, users now converse with their devices to accomplish many tasks previously done manually. Parsing task-oriented dialogs is the problem of understanding the user's intent and any associated arguments so that the assistant can fulfill the requested task.

Given the prominence of data-driven methods in NLP, public availability of relevant datasets dictates which problems can be effectively studied by the research community at large. Datasets such as MultiWoz (Budzianowski et al., 2018; Eric et al., 2020), TOP (Gupta et al., 2018), MTOP (Li et al., 2021), and SMCalFlow (Andreas et al., 2020) have enabled researchers to study numerous developments in conversational semantic parsing, e.g., Budzianowski and Vulić (2019), Pasupat et al. (2021). However, many challenging aspects of parsing, such as disfluencies, code-switching, and the use of the structured context which grounds conversations, have been missing in these datasets, stifling meaningful research progress in those areas.

With that in mind, we present PRESTO, a dataset that better represents real conversations users have with their virtual assistants. PRESTO consists of:

- Challenging conversational phenomena such as disfluencies, code-switching and user revisions, addressing some of the aforementioned limitations of current virtual assistants and enabling researchers to test new methods of addressing these challenges.

- Realistic multilingual dialogues. Unlike in other datasets, conversations in each language were not obtained by translating English dialog, but instead contributed by native speakers of each language.

- Human and synthetic structured context with the conversations. The data contributors were instructed to look at and optionally reference relevant context in the user's virtual environment. For example, contributors could reference the user's contact list while initiating a phone call or sending a message using the virtual assistant.

When users speak naturally to their virtual assistants, a variety of conversational phenomena are observed: disfluent turns (Gupta et al., 2021), revisions (also known as conversational repair (Cassell et al., 2000)) and code-switching (also known as

---

[*] Equal contribution.
[†] Work done while at Google.
[‡] Work done during an internship at Google.
[1] https://github.com/google-research-datasets/presto

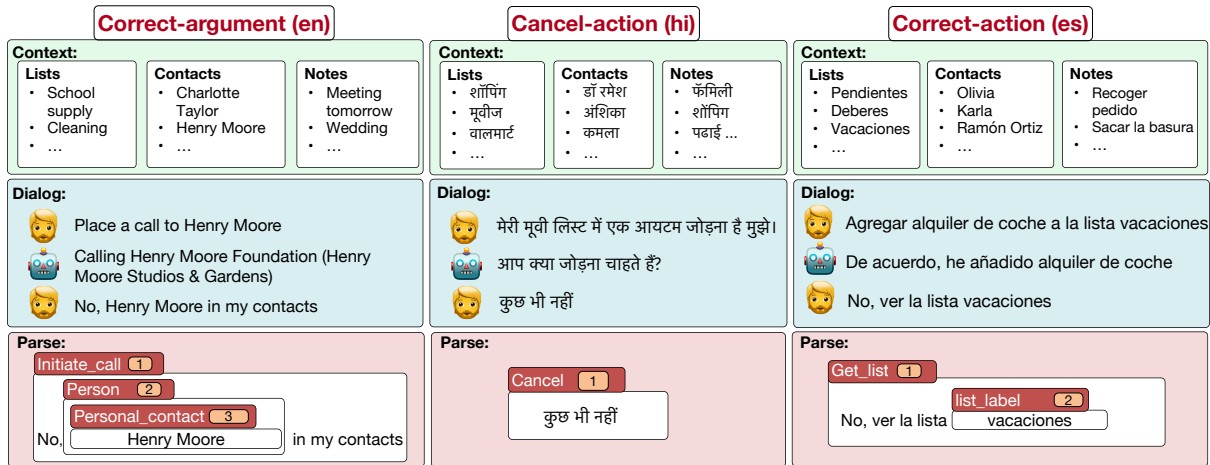

Figure 1: Examples of user revision dialog sessions from PRESTO. PRESTO includes annotated dialogs in 6 languages (*de, en, es, fr, hi, ja*) with various characteristics such as corrections (`correct-argument`, `correct-action`), cancellations (`cancel-action`) etc. Each example consists of Input: a user's virtual state (context), one or more user utterances and the corresponding virtual assistant responses (dialog). Output: the semantic parse of the last user utterance in the dialog (parse).

code-mixing (Agarwal et al., 2022)), to name a few. Our dataset highlights several such linguistic phenomena, and provides thousands (Table 1) of examples of each phenomenon per language, each produced by a native speaker. Overall, our dataset contains more than 550k annotated utterances across six languages. Each user utterance is further verified by three native speakers for fluency and correctness and then annotated separately twice. Section §3 discusses how the data was collected.

Section §2 discusses context as well as other important phenomena which are well-represented in this dataset such as user revisions and disfluent utterances. In section §4, we use mT5-based models (Xue et al., 2021) to present some baseline model performance metrics for this dataset, e.g., exact match accuracy on the test sets, the relative performance of monolingually- vs. multilingually-trained models, and data scaling effects with respect to various linguistic phenomena.

## 2 Dataset Characteristics

In this section, we discuss the characteristics of PRESTO dataset that set it apart from existing datasets: native conversations in multiple languages, code-switched utterances, user revisions, disfluencies, and structured context.

**Native Speakers.** The PRESTO dataset only includes utterances provided by native speakers of the language, with no translation. Table 1 shows the number of examples in each language.

Prior large multilingual datasets for conversational semantic parsing such as MTOP (Li et al., 2021) and MASSIVE (FitzGerald et al., 2022) contain non-English conversations obtained by translating English conversations to other languages,[2] resulting in *unnatural* and synthetic utterances which are unlikely to be spoken by native speakers of the non-English language. For example, an Arabic example in MASSIVE:

أحتاج إلى الحصول على خدمات الموقع هل
يمكنك التحقق

does not resemble native Arabic speech of any dialect. The corresponding English utterance in the MASSIVE dataset is: *I need to have location services on can you check.*

**Code-Switched Utterances.** Multilingual users often mix words from two languages in the same utterance, as shown in Fig. 2. Recognizing the difficulty of parsing such utterances, we asked bilingual data contributors to provide mixed utterances, and marked these examples with a special tag 'code-switching' to enable isolated research focused on addressing this phenomenon. Table 1 shows the number of code-switched examples in

---

[2]The translation was constrained in order to ensure the English source and the translation have the same semantic parse with argument values in English translated to the corresponding phrases in the other language.

| Language | # Intents / # Slots | Avg. Slots / Utterance | Avg. Tokens / Utterance | Avg. Prev. Utterances | Total Examples | Code Switching | User Revisions | Spoken Disfluencies |
|---|---|---|---|---|---|---|---|---|
| German | 34 / 285 | 1.60 | 9.17 | 2.56 | 83,584 | 12,357 | 22,129 | 16,781 |
| English | 34 / 303 | 1.57 | 9.03 | 2.49 | 95,671 | 5,918 | 27,741 | 17,588 |
| Spanish | 34 / 299 | 1.72 | 10.70 | 2.57 | 96,164 | 12,570 | 27,713 | 21,510 |
| French | 34 / 303 | 1.63 | 10.73 | 2.63 | 95,870 | 12,939 | 25,157 | 20,137 |
| Hindi | 34 / 285 | 1.47 | 9.32 | 2.55 | 72,107 | 16,517 | 15,833 | 10,193 |
| Japanese | 34 / 292 | 1.73 | 11.23 | 2.58 | 109,528 | 15,200 | 29,474 | 23,838 |
| Total | - - | 1.63 | 10.11 | 2.57 | 552,924 | 75,501 | 148,047 | 110,047 |

Table 1: Corpus and sentence level statistics of PRESTO, slices per language, including various linguistic phenomenon.

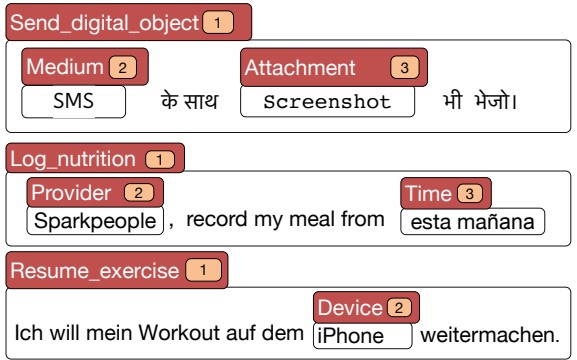

Figure 2: Examples of Hindi-English (*Send a screenshot along with the SMS.*), Spanish-English (*Sparkpeople, record my meal from this morning.*), and German-English (*I want to continue my workout on the IPhone.*) code-switched utterances from PRESTO.

each language.[3]

**User Revisions.** The example in Fig. 1 highlights another important aspect of realistic conversations: users often revise their request to the assistant. We have 4 such tags in our dataset which are all grouped under the broader user-revision linguistic phenomenon. The tags are: correct-action, correct-argument, within-turn-correction, and cancel- action. Sometimes the revision is necessary due to a mistake made by the virtual assistant, as in Fig. 1 (*correct-argument*). At other times, the user may simply change their mind about what they want the assistant to do which may happen in the same utterance, e.g., "*Add bread no no wait add wheat bread to my shopping list*", or in a later utterance, e.g., "*Sorry add wheat bread instead.*" Another common revision users make is to cancel their requests, e.g., "*Sorry don't add anything*". Fig. 1 shows examples of some of these revisions. Table 1 reflects the number of examples

with user revisions in the last utterance for each language in PRESTO.

**Disfluencies.** Due to the spoken nature of most conversations with virtual assistants, user utterances frequently have disfluencies such as repeated phrases and filler words, as shown in Fig. 3. Table 1 shows the number of examples with disfluencies for each language in PRESTO.

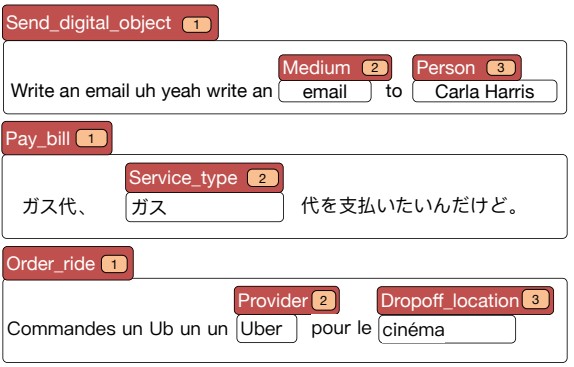

Figure 3: Examples of utterances in English, Japanese (*Gas bill, I want to pay the gas bill.*), and French (*Order an Ub an Uber for the cinema*) with disfluencies (filler words or repetitions) from PRESTO.

**Structured Context.** Users interact with virtual assistants in a virtual world (henceforth *context*) that may include structured objects like contacts, shopping lists, to-do lists, alarms, smart devices, etc. Context may or may not be needed to interpret user utterances. Models should be able to model and ignore structured information based on the query. In Fig. 1 (correct-argument), the virtual assistant fails to relate '*Henry Moore*' to the user's contacts and incorrectly interprets the name as a reference to the "*Henry Moore Foundation*". The gold parse provided in the dataset shows the correct interpretation for the last utterance, leveraging the structured context.

---

[3]We use a language ID classifier to determine which of the two languages is dominant in a given utterance.

| Dataset | # Languages | Multi-turn | Explicit Context | Labeled Conversational Phenomena |
|---|---|---|---|---|
| **PRESTO** | 6 | ✓ | ✓ | Code-Switching, Disfluencies, User-Revisions, Coreferences* |
| TreeDST (Cheng et al., 2020) | 1 | ✓ | ✗ | Coreferences |
| SMCalFlow (Andreas et al., 2020) | 1 | ✓ | ✗ | Coreferences, User-Revisions |
| MultiWOZ (Budzianowski et al., 2018) | 1 | ✓ | ✗ | Coreferences* |
| DSTC10 (Kim et al., 2022) | 1 | ✓ | ✗ | Disfluencies* |
| TOP (Gupta et al., 2018) | 1 | ✗ | ✗ | - |
| MTOP (Li et al., 2021) | 6 | ✗ | ✗ | - |
| MASSIVE (FitzGerald et al., 2022) | 51 | ✗ | ✗ | - |
| SNIPS (Coucke et al., 2018) | 1 | ✗ | ✗ | - |
| MultiATIS++ (Xu et al., 2020) | 9 | ✗ | ✗ | - |

Table 2: Comparison of PRESTO with other recent datasets on or related to semantic parsing of task-oriented dialog. * denotes phenomena which is implicitly present in the dataset but not explicitly tagged.

**Other Statistics.** PRESTO covers 34 intents, including the intent 'Other' which is a catch-all intent used for all out-of-scope utterances and constitutes 11.3% of all examples in the dataset. The average number of the previous user turns is around 2.6, which is representative of the relatively short conversations users tend to have with virtual assistants today. Table 1 provides summary statistics for the number of tokens in labeled user utterances in each language, demonstrating the large variety in utterance lengths.

**Related Datasets** There have been multiple related datasets which are listed in Table 2. The existing task-oriented dialog datasets can be split by capabilities and representations. The salient capabilities are – single turn or conversational, monolingual or multilingual, presence of conversational phenomenon, and single-domain vs multi-domain. In terms of representations, there are 2 popular paradigms, a direct semantic parse typically consisting of intents and slots vs maintaining an explicit dialog state. With the popularity of pretrained text-to-text models, the natural modeling choice for all of these has converged to similar models. In fact, our context representation can be seen as a constant dialog state for that conversation. While other datasets have focussed on multi-intent and compositional utterances (Cheng et al., 2020), PRESTO emphasizes on labelled conversational phenomena and explicit context, which are important for modeling real-world task-oriented dialogues.

## 3 Data Collection

The scope for data collection is based on the App Actions built-in intents available for 3rd-party An-

droid Developers, which cover a wide spectrum of domains including finance, health, fitness, food, transportation, social, communications, shopping among others.[4] Our dataset has over 34 unique intents and 370 arguments across 6 languages. Table 6 talks about the intent distribution in the dataset. We use two complementary approaches to collect contextual and non-contextual examples.

**Contextual Examples.** Data contributors were asked to have conversations with a virtual assistant simulator based on different sets of instructions for each language and for each type of targeted linguistic phenomenon (e.g., code-switching). Each data collection request targeted a single intent and was initialized with potentially relevant structured context, e.g., contacts, notes or lists, which can optionally be referenced by data contributors in the utterances they produce. The initial seed contacts, notes and lists are authored by native speakers and are shared for all examples in one request. For example, when initiating a call, data contributors may use one of the names in the contacts (e.g., "*call mom*"), but they may also use the name of a business not in the contact list (e.g., "*call mcdonalds*").

Each user utterance from the collected conversations is exported as a candidate example for semantic annotation by trained expert linguists who produce two types of annotations.[5] First, linguists decide whether the example is in scope or not. Out of scope examples include incoherent or nonsen-

---

[4]https://developer.android.com/reference/app-actions/built-in-intents

[5]Due to limited annotation capacity, we were not able to annotate all candidate examples and we prioritized candidate examples which have at least one previous turn. Candidates which were not included were discarded and are not included in this data release.

sical utterances, unsupported arguments and out-of-scope intents. Then, they choose the intent and nested arguments as shown in Fig. 1. Finally, they added semantic tags which indicate which linguistic phenomena of interest are expressed in this example, such as, `correct-argument`, `cancel-action`, `code-switching`, etc.

**Non-Contextual Examples.** A key challenge in representing context in real world semantic parsing is that most examples can be parsed without using it, for example utterances like "*play music*" are typically non-contextual but an utterance like "*play workout music*" might be contextual if the user has a playlist called "*workout*". This makes it a challenging modeling task; context ideally should only be used when it is relevant. On the one hand, exclusively using relevant contextual examples when training a conversational semantic parser results in models which are over-sensitive to irrelevant contextual features. On the other hand, exclusively using non-contextual training examples where the user utterance is not related to previous turns or structured context teaches the model that the context does not matter, and may result in emphasizing arbitrary contextual features which happen to correlate with some prediction in the training data.

A balanced approach is to include both contextual and non-contextual examples in training and use targeted evaluation sets to understand the impact of different ways of representing context. We obtain non-contextual examples by asking linguists to produce single-turn utterances along with their gold parses. Then, we pair them with samples of the structured context and previous turns from the contextual conversations (described later in the post-processing section). We denote such synthetic generated context in the data by the tag `context:synthetic`.

**Data Quality.** We adopt several mechanisms for boosting the data quality of collected data. All our data collection is done in 2 phases. Phase 1 is data collection, where annotators come up with queries pertaining to a scenario. Phase 2 is the actual semantic parse annotation for the query.

For non-contextual examples, we use a crowd compute like platform with native speakers to generate and annotate the data. The process is described below:

1. **[Phase 1.1]** Annotator 1 authors a query (contextual or non-contextual).

2. **[Phase 1.2]** Annotators 2, 3, 4 independently validate the query (scope judgment). If all 2, 3, 4 agree that the query is valid, then the query goes to phase 2. Upon any disagreement, the example is discarded.

3. **[Phase 2.1]** Annotators 5 and 6 annotate the query independently. If their annotations match, the query is considered resolved along with the annotation.

4. **[Phase 2.2]** If 5, 6 disagree, the query is sent to annotator 7 who annotates the query. If 7 matches any of 5 or 6, then the query is resolved to the matching annotation else the example is discarded.

As can be seen, the overall query generation and annotation process is similar for contextual and non-contextual examples. However, contextual examples require more care in annotation therefore we additionally ask trained in-house linguists who also speak the language to verify the plausibility of each example and annotations and exclude ones that are out of scope or unclear as discussed earlier.

We manually analyzed 6K examples to verify the efficacy of our approach. We assessed the correctness of the parse and the plausibility of the user utterance. We used 3 labels to assess the semantic parse quality: '`Accuracy:GOOD`', '`Accuracy:BAD:INTENT`', '`Accuracy:BAD:SLOT`'. We used 2 labels to assess the utterance: '`Acceptability:HIGH`' and '`Acceptability:LOW`'.

'`Accuracy:GOOD`' is used when the top-level intent and all slot values are accurate. '`Accuracy:BAD:INTENT`' is used when an intent is not accurate. '`Accuracy:BAD:SLOT`' is used when the intent is accurate but one or more slot values are not accurate. '`Acceptability:HIGH`' is used when the utterance is likely to be spoken or typed by Assistant users, and is grammatically and contextually correct, otherwise we use `Acceptability:BAD`.

Before post-processing, annotators agreed on utterance plausibility (acceptability) for 92.9% of examples, and agreed on full semantic parse annotation (accuracy) for 90.7% of examples.

The breakdown of intent and argument errors in contextual examples can be found in Table 3 for five of the six languages (Spanish was not included in this analysis due to capacity constraints).

| Language | Bad Intent | Bad Argument |
|----------|-----------|--------------|
| German | 2.8% | 3.7% |
| English | 2.9% | 8.5% |
| French | 0.0% | 2.5% |
| Hindi | 1.9% | 4.8% |
| Japanese | 4.3% | 2.1% |

Table 3: Type of accuracy errors in examples by language from our human evaluation.

**Post-processing.** One of the common annotation errors we found is that code-switched examples often had the wrong language ID (`LangID`) associated with them. To address this, we use a language ID classifier to determine the dominant language in code-switched examples.

Some arguments (due to fine-grained domain modeling) were often confused for other arguments that have similar (but distinct) semantics, e.g., `ListItem` vs. `ExistingListItem`, resulting in inconsistent annotations and contributing to the error rate discussed earlier. To address this, we merged pairs of argument names where the distinction rarely matters in practice. To make our data uniform, we add synthetic context and previous turns to non-contextual examples. Examples that are missing previous turns are augmented to reuse previous turns from other examples by first sampling the number of previous turns using a negative binomial distribution with $r = 5, p = 0.8$, then sampling one of the collected conversations that has this many turns. Examples that are missing a given type of structured context, e.g., contacts, are augmented by randomly sampling contact names from related argument values specified by our data contributors, which results in realistic and diverse contexts.

We do another quality check to measure Accuracy and Acceptability after post-processing by randomly sampling 500 queries from each language (except Spanish due to capacity constraints). We ask 3 independent annotators (same as the earlier quality check) to rate the queries. After post-processing, we find that linguists agree, defined by 2 out of 3 annotators independently agreeing, on utterance plausibility (acceptability) for **98.8%** of examples and agree on full semantic parse annotation (accuracy) for **93.24%** of examples. This shows our post-processing decreased the data noise significantly.

## 4 Experiments

In this section, we demonstrate a few experimental setups which are enabled by PRESTO, provide baseline results for future work, and summarize our findings.

### 4.1 Setup

Each example in the dataset is designated as train, development or test with respective probabilities 0.50, 0.15 and 0.35, which enables us to have large test sets even when doing focused evaluation on a particular phenomenon of interest as in §4.3. All data splits are provided as part of the data release. We use the `t5x` library (Roberts et al., 2022) to fine-tune mT5's public checkpoint (Xue et al., 2021) on the train portion of the data (unless otherwise stated in k-shot experiments).

For all experiments, we report **exact match accuracy** of the predicted semantic parse, which gives the model one point for each example where the intent and all the arguments are correctly predicted and zero points otherwise. All experiments are based on the mT5-Base model (580M parameters),[6] except the scaling experiments in §A.1 which demonstrate the effect of model scaling to mT5-Large (1.2B parameters), mT5-XL (3.7B parameters), and mT5-XXL (13B parameters). All models (except the monolingual models discussed in §4.5) are fine-tuned on the union of training examples from all languages in PRESTO. Few shot experiments for code switching, user revisions and disfluencies all share the same training set, e.g., the 5-shot model across all three sections consist of 5 code-switching examples, 5 cancellations, 5 within-turn corrections, 5 intent cross-turn corrections, 5 argument cross-turn corrections and 5 disfluency examples, in addition to all examples which do not represent any of these phenomena.

**Features.** The input sequence fed to the model consists of the last user utterance which needs to be parsed, followed by previous turns (both user and assistant turns) in reverse chronological order, followed by some representation of the structured context (more on this in §4.4), with a separator token between consecutive fields. An example is shown in Figure 4.

**Hyper-parameters** We do a rough hyperparameter tuning and fine-tune the model for 20K steps.

---

[6] https://github.com/google-research/t5x/blob/main/t5x/examples/t5/mt5/base.gin

**Input:**

```
No, Henry Moore in my contacts | Calling Henry
Moore Foundation (Henry Moore Studios & Gardens)
| Place a call to Henry Moore [SEP] Lists: School
Supply, Cleaning, ... [SEP] Contacts: Charlotte
Taylor, Henry Moore, ... [SEP] Notes: Meeting
tomorrow, ...
```

**Output:**

```
Initiate_call( callee = Personal_contact( person
= Henry Moore ) )
```

Figure 4: Example serialization and output of features. The top box shows the encoding for the first input in Fig. 1. The output sequence is a depth-first traversal of the semantic parse.

We use the Adafactor optimizer with 0.8 decay rate and 0 step offset. We have a max input length of 512 for training and 1024 for inference and a batch size of 128.

## 4.2 Overall Results

Before diving into focused evaluations on various phenomena of interest, it is instructive to examine the overall model performance on this dataset when trained on 100, 1K, 10K examples, as well as the full the training set. The results are shown in Fig. 5, and they demonstrate a linear increase in model performance as the number of training examples grows exponentially.

We next assess how difficult it is for mT5-based models to parse examples with different linguistic phenomena. Table 4 shows the exact match accuracy for a multilingual model trained on examples without these phenomena. Each column corresponds to a different test set with examples that have user revisions, disfluent utterances, code-switching, or none of the above. Unsurprisingly, the models perform much worse on examples with these phenomena when they are not exposed to them in training. In contrast, when the model is exposed to a lot of training examples with these phenomena, as in Table 7, the performance gap across the different test sets is smaller, except for code-switching.

## 4.3 Results on Linguistic Phenomena

We evaluated how well an LLM-based model parses examples with different linguistic phenomena when fine-tuned on a large number of multilingual examples but only a limited number of

examples with the relevant phenomena. We trained several mT5-base models on different training sets with varying numbers of examples with each linguistic phenomenon: 0, 5, 25, 125, 625, 3125, and 15,625 examples [7]. We then evaluated the models on a test set with only examples of the relevant phenomena.

**Code-Switching** As seen in Fig. 5, when no code-switching examples were used in training, the model's exact match accuracy ranged between 56-72%, performing best on Japanese and worst on Hindi. Even at 0 shot, we saw a relatively high performance for this phenomenon. We hypothesize that this is due to using a LangID to identify code-switching utterances, which may cause code-switched utterances to leak into the training set.

**User revisions** Similar to code-switching, this setting has the training dominated by examples without user revisions, while the test set only included examples with user revisions. The zero-shot exact match accuracy ranged between 18-28%. Unlike code switching, adding only 25 examples of each type of user revisions notably improved the performance on user revisions. After the first few examples, the performance improved linearly as the number of examples with relevant phenomena grew exponentially from 25 to 125 to 625. The performance slowed down when the number of examples reached 15K.

**Disfluencies** Here, we turn to disfluent utterances and try to estimate how well our models fare on them with various number of disfluent examples in the training set. Fig. 5 shows the results for the focused test set with disfluent utterances. In the zero-shot setup, the performance starts at a higher accuracy range for all languages, compared to use revisions. As the number of training examples grows exponentially, we observe a linear increase in exact match accuracy for all languages.

## 4.4 Structured Context

PRESTO provides an opportunity to examine how structured context can be used to improve conversational parsing models. We use a focused test set that highlights context by only including contextual examples (first approach in §3)).

---

[7]For user revisions, 5 examples would mean 5 examples with intent corrections + 5 with argument corrections + 5 with within-turn corrections + 5 with cancellations

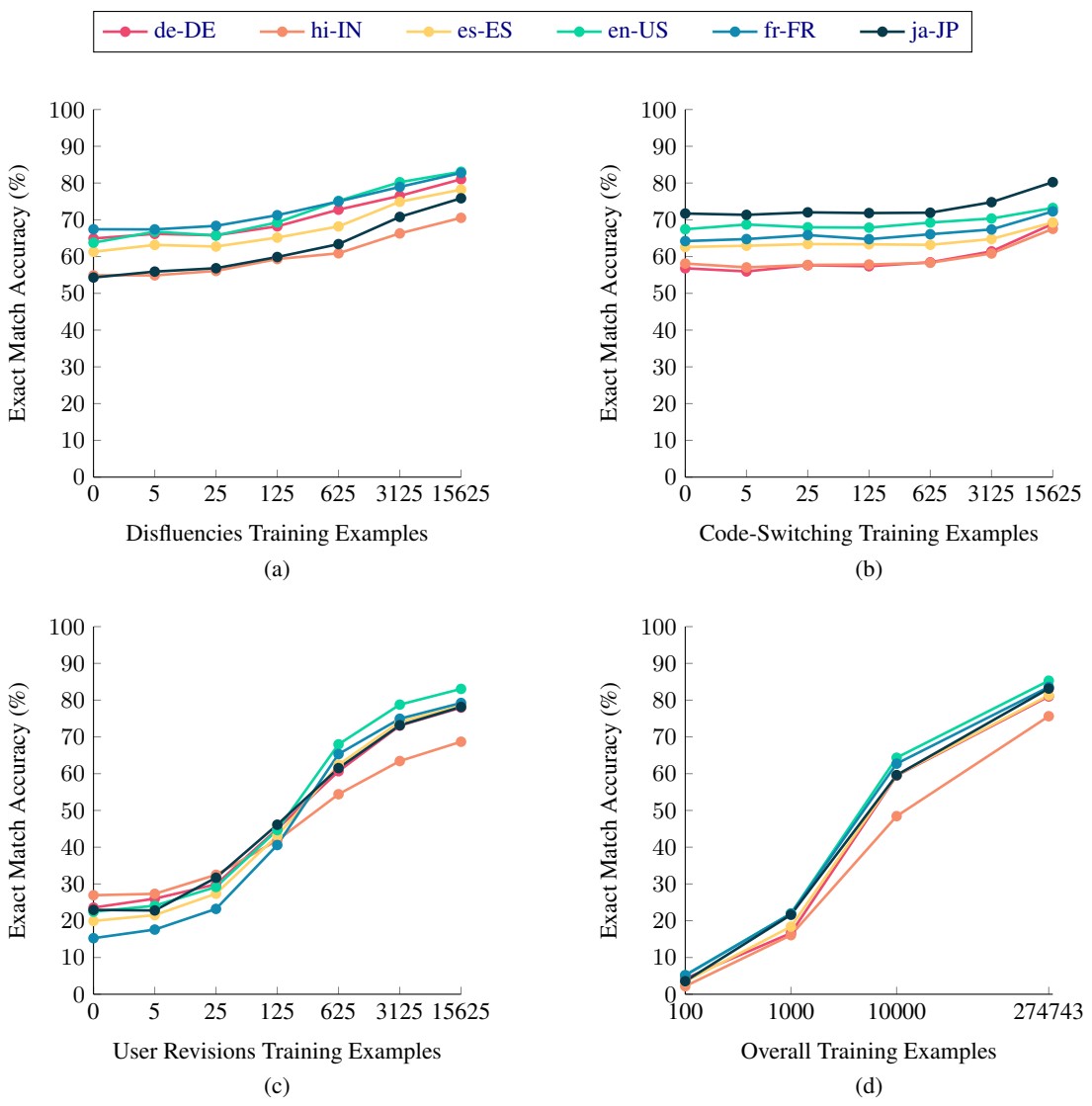

Figure 5: Scaling plots of the baseline multilingual mT5 model on test set of PRESTO for: (a) disfluencies, (b) code-mixing, (c) user revisions, and (d) all inclusive (i.e. sampling uniformly from the overall PRESTO training data mixture).

We linearize and prepend the context tokens to the T5 parser input with [SEP] tokens as shown in the example presented in 4.1. Initially, we simply linearized the entire context and added that as a feature. We noticed that naively prepending all the context had too much noise and was not resulting in performance improvements. Hence to represent context in the input sequence, we first identified context notes, contact names and list names which had a trigram similarity greater than 0.6 with the last user utterance and only used the last two turns. We call this 'filtered context representation' since it filters out parts of the context which are unlikely to provide useful information for the parser.

First, we examine how the performance increases on the focused test set as we include more contextual examples to the training set. In particular, we fine-tune the mT5-Base model with filtered context representation on all the non-contextual training examples in addition to 10, 100, 1K or 10K contextual training examples. Table 5 provides average performance for 5 languages with and without context. We notice very little improvement with addition of the context features. Our interpretation for this result is that, even though the data contributors were encouraged to use the structured context when conversing with the virtual assistant, most utterances can be understood without referencing the structured context. For example, it is easy for the model to parse the utterance "add apples to my shopping list", with no knowledge about what lists the user have.(See §3 for more details on contextual

| Language | No Phenomenon | User Revisions | Disfluency | Code-Switching |
|---|---|---|---|---|
| German | 81.76 | 23.56 | 64.91 | 56.81 |
| English | 85.16 | 22.45 | 63.78 | 67.43 |
| Spanish | 81.26 | 19.94 | 61.31 | 62.60 |
| Japanese | 83.41 | 22.95 | 54.31 | 71.71 |
| Hindi | 76.60 | 26.94 | 54.89 | 58.10 |
| French | 84.27 | 15.25 | 67.42 | 64.21 |
| Overall | 82.38 | 21.51 | 61.26 | 63.16 |

Table 4: Exact match accuracy results (%) on the test set for the zero-shot multilingual mT5 model (i.e. trained on all examples with no marked phenomena).

vs. non-contextual examples.)

| Number of Examples | Non-Contextual Model | Contextual Model |
|---|---|---|
| 10 | 21.23 | 23.94 |
| 100 | 40.84 | 41.90 |
| 1000 | 64.29 | 64.93 |
| 10000 | 80.28 | 80.50 |

Table 5: Average performance (exact match accuracy (%)) on the contextual test set across all the languages for contextual vs non contextual models. It can be seen that in low resource setup contextual models outperform non-contextual model, i.e. modeling context helps.

## 4.5 Monolingual vs. Multilingual Models

In previous sections, we train a single parser on training examples from all languages in PRESTO, as proposed in Ammar et al. (2016). A more traditional approach is to train several monolingual models. Figure 6 shows how the monolingual models compare to multilingual models with varying amounts of training data. Our results show that in lower data regimes there is a clear gap between monolingual and multilingual models, but when using all training instances in the dataset, monolingual and multilingual models converge towards similar performance, in terms of overall accuracy.

## 5 Conclusion

We introduce PRESTO, a over 550K-examples, six-language dataset for parsing realistic task-oriented dialogs. PRESTO is enriched with a high representation of contextual examples, code switching, user revisions and disfluencies. Our initial results demonstrate that it is possible for our models to perform well on various linguistic phenomena, but only when the model is exposed to a very large number of training examples with the given phenomenon. We observe that feeding a simple representation of the structured context to the model

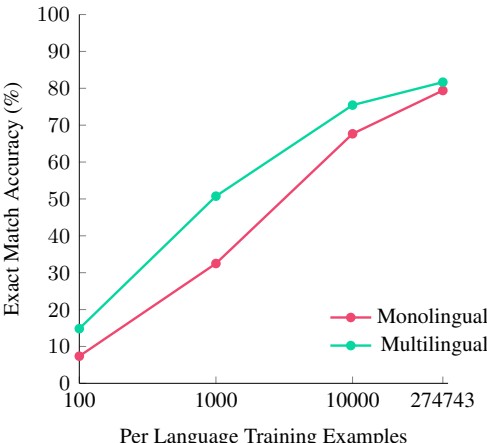

Figure 6: Average test set performance of multilingual and monolingual models. In low resource setting, we can see that a multilingual model performs better than a monolingual one.

does not yield large improvements on contextual examples. We also notice that multilingual models perform better than monolingual models in lower data regimes.

**Open Questions.** By introducing a new dataset with unique characteristics, this work deliberately asks more questions than it answers. Some of the obvious modeling questions which this dataset can help answer are: How do we train models which perform well on conversational phenomenon like code-switching, user revisions and disfluencies, with a relatively small number of training examples that exhibit these phenomena? How do we identify examples for which context is critical to understanding the user's utterance? How can structured context be represented in our models most optimally? How can we increase cross-lingual supervision in multilingual models to make them more data efficient? We hope that PRESTO is useful in answering these and other questions about conversational semantic parsing.

## Limitations

The PRESTO dataset was collected before the release of the ChatGPT API[8], resulting in the absence of any performance testing on the newly collected phenomena during data collection. Furthermore, the extensive size of the dataset poses challenges in testing models such as ChatGPT on PRESTO without incurring supplementary expenses. Pan et al. (2023) employed a multi-turn prompt framework to evaluate ChatGPT's performance on comparable datasets and reported a Joint Goal Accuracy (JGA) of 64.23 on MultiWOZ2.4, compared to the state-of-the-art (SOTA) finetuned score of 75.90. Table 2 demonstrates that MultiWOZ is less intricate than PRESTO. Thus, we consider the PRESTO dataset to remain a challenging benchmark for such models. Subsequently, future research can focus on conducting additional modeling experiments.

## Ethical Considerations

The PRESTO dataset does not incorporate copyrighted material or otherwise violate intellectual property of third parties since all the raw text was directly collected from trained data contributors who received fair market compensation for their work. All data contributors were vetted to be native speakers for the contributed data, although their nationalities may vary. See Section 3 for more details on data collection, and Section 2 for more details on data characteristics.

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

## A  Appendix

### A.1  Model Scaling

We present the results on the overall dataset as well as the model scaling results in this section. Table 7 presents results on the test sets if we use all the available training data.

The experiments presented so far use mT5-Base since it incurs lower cost than larger models and provides reasonable performance on many NLP tasks. In this section, we examine larger model sizes impact the results on this dataset. As a case study, we focus on the 125-shot models for code switching, disfluencies and user revisions. Fig. 7 demonstrates the impact of increased model size on the overall test results. As expected, the larger models consistently give better results but are still lower than the best results we were able to achieve by using the full training set with mT5-Base (discussed earlier in Fig. 5). Fig. 7 demonstrates a similar pattern for the focused evaluation on user revisions, and we see a similar pattern for other phenomena. These results confirm that mT5-Base strikes a good balance between model size and performance, and we recommend using mT5-Base for extensions of this work.

| Intent | German | English | Spanish | French | Hindi | Japanese |
|---|---|---|---|---|---|---|
| Add_contact | 2779 | 2931 | 3382 | 3148 | 2122 | 3527 |
| Add_item_to_list | 944 | 2959 | 2913 | 986 | 1958 | 2503 |
| BuyEventTickets | 2550 | 2463 | 2883 | 2842 | 1697 | 3049 |
| Cancel | 1479 | 3029 | 2061 | 1775 | 2318 | 2525 |
| Cancel_ride | 2252 | 2353 | 3239 | 3047 | 2053 | 3098 |
| Check_order_status | 2700 | 2877 | 3448 | 3455 | 2136 | 3934 |
| Create_list | 844 | 594 | 745 | 1063 | 778 | 1023 |
| Create_note | 1566 | 1700 | 1903 | 1819 | 1568 | 1998 |
| Find_parking | 4562 | 1210 | 4387 | 2345 | 3117 | 2692 |
| GetGenericBusinessType | 3130 | 2613 | 2782 | 2904 | 1676 | 3139 |
| Get_bill | 2249 | 2466 | 2922 | 2582 | 1857 | 3275 |
| Get_health_stats | 2477 | 2512 | 2827 | 2847 | 1982 | 2879 |
| Get_list | 701 | 705 | 698 | 676 | 856 | 761 |
| Get_message_content | 2102 | 2142 | 2371 | 2343 | 1320 | 2879 |
| Get_note | 1737 | 1905 | 2022 | 2277 | 1978 | 2138 |
| Get_product | 2341 | 2076 | 2483 | 2438 | 1574 | 3344 |
| Get_security_price | 2175 | 2333 | 2707 | 2425 | 1752 | 3245 |
| Initiate_call | 1822 | 4292 | 2239 | 1965 | 1807 | 2250 |
| Log_exercise | 2602 | 2708 | 2050 | 2663 | 1962 | 3049 |
| Log_nutrition | 2219 | 2459 | 2667 | 2605 | 1786 | 3204 |
| Open_app | 2488 | 2497 | 3367 | 3076 | 2069 | 3643 |
| Order_menu_item | 224 | 860 | 659 | 736 | 155 | 740 |
| Order_ride | 2175 | 2342 | 2857 | 2992 | 1809 | 2993 |
| Other | 8457 | 17499 | 5650 | 10937 | 9163 | 13737 |
| Pause_exercise | 2821 | 2995 | 3109 | 3149 | 1830 | 3495 |
| Pay_bill | 2280 | 2304 | 2837 | 2823 | 1834 | 3048 |
| Play_game | 2640 | 2667 | 2889 | 3048 | 1941 | 3074 |
| Post_message | 2597 | 2224 | 3065 | 3278 | 2049 | 3529 |
| Record_video | 2609 | 2295 | 3317 | 3369 | 1934 | 3864 |
| Resume_exercise | 2367 | 2060 | 3136 | 2929 | 1991 | 2896 |
| Send_digital_object | 2307 | 3127 | 2804 | 2652 | 2016 | 2905 |
| Start_exercise | 2719 | 3030 | 3272 | 3182 | 2031 | 3311 |
| Stop_exercise | 2719 | 2814 | 3207 | 3127 | 2078 | 3432 |
| Take_photo | 3950 | 2630 | 5266 | 4367 | 4910 | 4349 |
| Overall | 83584 | 95671 | 96164 | 95870 | 72107 | 109528 |

Table 6: Intent distribution by language for PRESTO.

| Language | No Phenomenon | User Revisions | Disfluency | Code-Switching |
|---|---|---|---|---|
| German | 82.41 | 81.33 | 84.19 | 72.50 |
| English | 86.18 | 85.36 | 86.20 | 75.99 |
| Spanish | 82.95 | 82.26 | 82.81 | 72.22 |
| Japanese | 84.91 | 82.22 | 80.46 | 84.41 |
| Hindi | 78.30 | 73.24 | 75.49 | 73.24 |
| French | 85.42 | 82.99 | 85.82 | 75.98 |
| Overall | 83.66 | 81.85 | 82.92 | 75.88 |

Table 7: Exact match accuracy results (%) for the multilingual mT5 model trained on the full training set of PRESTO.

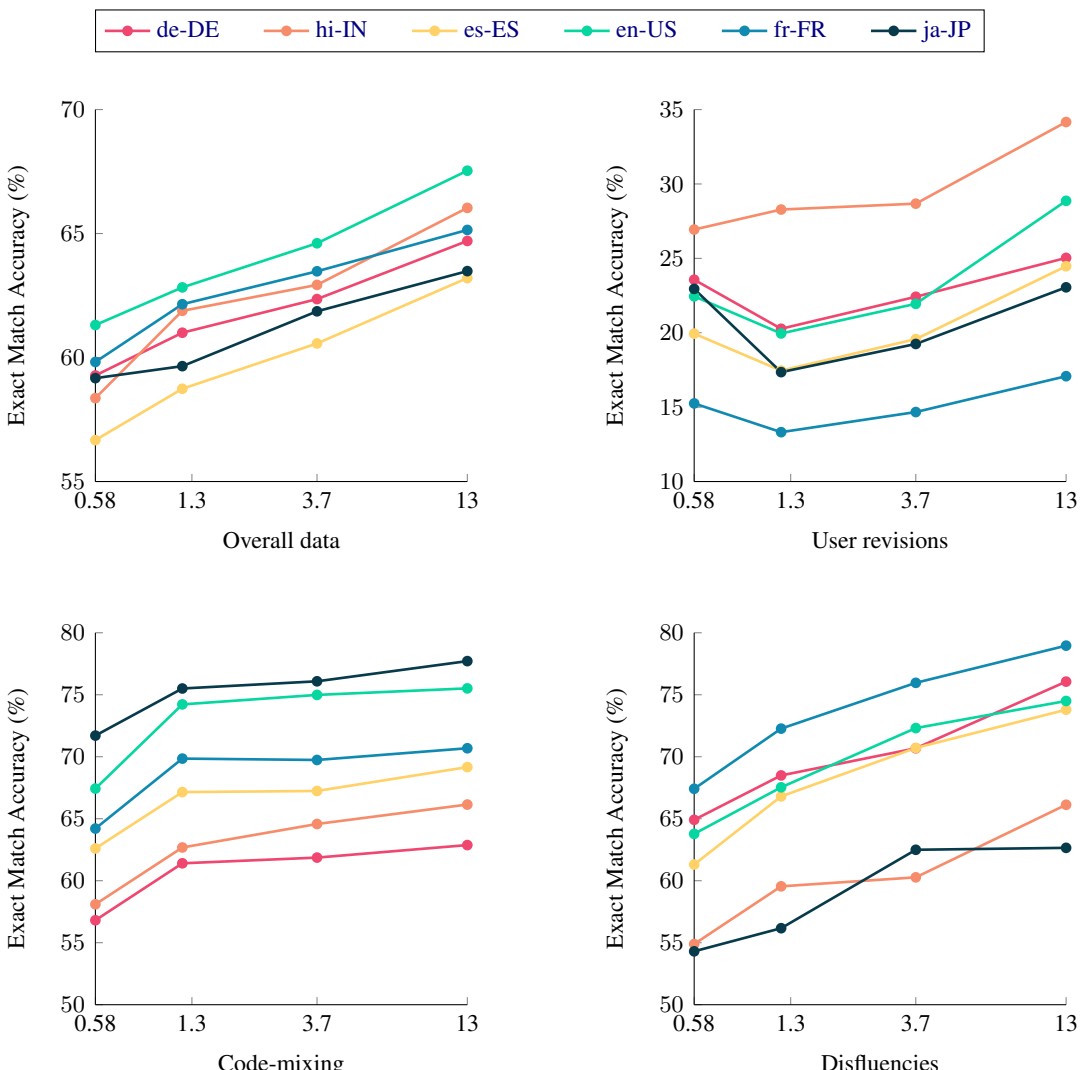

Figure 7: Scaling plots for increasing the model capacity. This scaling study was done using 125 shot training set. As expected larger models generally perform better.