# OpenReview forum: "PRESTO: A Multilingual Dataset for Parsing Realistic Task-Oriented Dialogs"
_EMNLP/2023/Conference — EMNLP 2023 Main_

### Official Review · Reviewer_8Nj3 · 2023-08-03

**Typos Grammar Style And Presentation Improvements:** Section 4.3 is missing a discussion r…
**Soundness:** 4

**Excitement:**

4: Strong: This paper deepens the understanding of some phenomenon or lowers the barriers to an existing research direction.

**Paper Topic And Main Contributions:**

This paper presents a new multilingual dataset of conversations between humans and virtual assistants. It includes more than 550k annotated user utterances. The utterances are all by native speakers and cover six different languages of multiple families. Each conversation is also paired with context information (e.g. contacts, notes, lists), which can be referenced by the user. In addition to intents and slots, each utterance is also annotated for disfluencies, code switching, and user revisions. The data collection and annotation included several annotation and filtering steps to improve the quality of the dataset and its annotations. Preliminary experiments using fine-tuned versions of the multilingual T5 model show that these models can be robust to the multiple linguistic phenomena that occur in the dataset, but only when exposed to a large amount of training examples featuring those phenomena.

**Questions For The Authors:**

A. Does the dataset include audio? (I'm not completely sure whether the conversations were typed or transcribed)

**Reasons To Accept:**

Large annotated resources are always relevant for research. Furthermore, in this case, the dataset focuses on several linguistic phenomena that are not typically considered in previously existing datasets with conversations between humans and virtual assistants. Thus, it is particularly relevant, especially considering the multiple languages covered in the dataset. Finally, considering that data collection and annotation included multiple filtering steps to ensure quality, it is expected that the dataset is actually representative of the phenomena that occur in interactions between humans and virtual assistants.

**Reasons To Reject:**

I don't see any major reason to reject the paper.

**Reproducibility:**

4: Could mostly reproduce the results, but there may be some variation because of sample variance or minor variations in their interpretation of the protocol or method.

**Reviewer Confidence:**

4: Quite sure. I tried to check the important points carefully. It's unlikely, though conceivable, that I missed something that should affect my ratings.

---

> ### Author Rebuttal · Authors · 2023-08-28
>
> Thank you for your comments on our work. Our dataset does not include audio. The conversations were typed by the annotators. We will clarify this in the revised version. Due to limited space, the section on the influence of disfluencies in 4.3 was omitted. We will add it in the camera-ready version of the paper.

---

### Official Review · Reviewer_ePrg · 2023-08-05

**Soundness:** 4

**Excitement:**

4: Strong: This paper deepens the understanding of some phenomenon or lowers the barriers to an existing research direction.

**Paper Topic And Main Contributions:**

This paper presents PRESTO, a large (550k example) dataset for conversational semantic parsing. The dataset covers multiple languages and conversational phenomena, inspired by the kind of requests made by users of real conversational systems. The first half of the paper describes the design of the dataset and the annotation process; the second half demonstrates how the dataset can be used in experiments to shed light on what makes it challenging to build accurate natural language understanding systems.

**Questions For The Authors:**

A: For the code-switching examples, is English always one of the languages? Or can there be e.g. Spanish-French code-switching?

B:  On p. 5 the "Post-processing" step is found to improve acceptability (92.9%->98.8%) and accuracy (90.7%->93.24%). It would be helpful to understand which actions contributed to the improvements. The two actions listed in the main paper are relabelling for language ID and adding past turns for non-contextual examples; the appendix also mentions that confusable labels were merged (this should be in the main section too) - it's not immediately clear why any of these should improve acceptability of the user utterance since they do not change the query text?

C: On p.8 the authors state "most utterances can be understood without referencing the structured context". This is likely less true for more ambiguous utterances, e.g. "show Bob" rather than "show Bob in my contacts" or "show my note called Bob". Are such ambiguities captured in the dataset?

D: As shown in Figure 4, a turn contains the conversational history of user and system utterances. What is the space of possible system utterances? Does the system sometimes ask questions to prompt, confirm, disambiguate? Does this influence the interpretation of the target turn?

**Reasons To Accept:**

- As in many areas, progress in conversational system research is constrained by the availability of large realistic datasets. Since PRESTO contains some phenomena not present in other datasets (disfluencies, code-switching) and is larger than most, I expect it to become a widely-used benchmark dataset.

- The design of the data, including the focus on phenomena like disfluencies and structured context, is clearly informed by the experience of delivering language understanding systems to real users of virtual assistants. This perspective is of great value to the broader research community, including academic researchers who may have less exposure to the practical aspects of building high-quality assistants at scale.

- The paper is very clearly written.

- The experiments make sense and show meaningful insights into the task. E.g. the authors show that models trained on data without disfluencies find it difficult to understand data containing them.


**Reasons To Reject:**

No significant weaknesses.

**Reproducibility:**

3: Could reproduce the results with some difficulty. The settings of parameters are underspecified or subjectively determined; the training/evaluation data are not widely available.

**Reviewer Confidence:**

5: Positive that my evaluation is correct. I read the paper very carefully and I am very familiar with related work.

**Typos Grammar Style And Presentation Improvements:**

- line 168: I don't follow how Table 1 (which reports averages and totals rather than ranges and variances) demonstrates "the large variety in utterance lengths"

- p.4, "Contextual Examples": in the first step data contributors were told which targeted phenomenon (phenomena?) to include, e.g. code switching. Then linguists were asked to label annotations with "semantic tags which indicate which linguistic phenomena of interest are expressed in this example". If we knew already which phenomena are included, why do we need to check again - is it to verify that the data  contributors followed the instruction, or to label each turn inside a conversation?

- p.4, "Contextual Examples": these examples originate in full conversations and are then split up into utterances. Is it possible to group utterances and recover the original conversation sequence?

- p.4, "Non-Contextual Examples": to create conversations the authors concatenate a single-turn utterance with a set of previous utterances and structured context. It is possible that the presence of such context could change the interpretation of the single-turn utterance. Was any checking done to prevent this?

- Table 2 shows that PRESTO has features that other datasets do not. For transparency it would be good to mention features that other datasets have but PRESTO does not. For example, TreeDST contains multi-intent conversations and compositional utterances - if I understand correctly (p. 4, "each data collection request targeted a single intent") such phenomena are not included in PRESTO. I'd also suggest including the size of the various datasets in turns and/or conversations.

---

> ### Author Rebuttal · Authors · 2023-08-28
>
> Thank you for your comments and questions. Below are the answers to your questions:
>
>  - A) Yes. English is always one of the languages used in code-switched utterances. For this task, we selected annotators who were fluent in both English and the local language with which code-switching occurs.
>
>
>  - B) We will add a discussion on this in the revised version and will also include the discussion on merging the labels in the main section when given the extra page in the camera-ready version.
>
>
>  - C) In the annotation process, annotators were instructed to create queries based on the structured context. As a result, we have queries that refer to the context. However, it is difficult to determine how often ambiguity arises in parsing the query due to the context alone by simply examining the data. Our results suggest that most of the time, the utterance can be parsed without referencing the context.
>
>
>  - D) The system does ask clarifying questions to confirm and disambiguate user queries (an example is Figure 1 (cancel-action)). Our annotators were asked to continue the conversation with the assistant until they had reached a targeted intent in each of the data collection requests. We consider the entire context to influence the interpretation of the target turn and hence all ‘previous_turns’ are used in the input to predict the output.
>
> Other comments:
>  - Average utterances: Thank you for pointing that out. We will add the variances in the revised version to make that point clear.
>
>  - p4 contextual examples: Yes. We provide the entire sequence of utterances in a metadata field called ‘previous_turns’, so it is indeed possible to recover the original conversation sequence. We use the entire conversation in the input as seen in Figure 4.
>
>  - p4, Non-contextual examples: We computed the accuracy metric on a sample after post-processing to see if the system response still remains valid. As mentioned in the the post-processing section, our numbers indicate that this is not an issue.
>
>  - Indeed, PRESTO does not contain multi-intent conversations. We will add the relevant suggested information in the revised version of the paper.
>
> We thank you for your important questions and suggestions. We will take these points into consideration when writing our revised version of the paper with an extra page.

---

### Official Review · Reviewer_WWoB · 2023-08-07

**Soundness:** 4

**Excitement:**

3: Ambivalent: It has merits (e.g., it reports state-of-the-art results, the idea is nice), but there are key weaknesses (e.g., it describes incremental work), and it can significantly benefit from another round of revision. However, I won't object to accepting it if my co-reviewers champion it.

**Paper Topic And Main Contributions:**

This paper introduces a public dataset PRESTO, with over 550K contextual multilingual conversations between humans and virtual assistants. A series of experiments are performed with mT5-based baseline models.

**Reasons To Accept:**

1)	The paper's main contribution is its inclusion of diverse challenges found in real-world TOD tasks, such as disfluencies, revisions, and code-switching for multilingual tasks. This enhances the realism of the TOD dataset, which is often lacking in existing datasets.
2)	The experiment setup is commendable, incorporating exact match accuracy evaluation, diverse training sets with varying example numbers for each linguistic phenomenon, and comparisons between contextual and non-contextual models, as well as monolingual and multilingual models. These choices lead to several interesting findings discussed in the conclusion.


**Reasons To Reject:**

The baseline model used in this paper appears to be weak, as all experiments solely rely on fine-tuning the mT5 model without any comparisons to other SOTA models in the TOD domain.

**Reproducibility:**

4: Could mostly reproduce the results, but there may be some variation because of sample variance or minor variations in their interpretation of the protocol or method.

**Reviewer Confidence:**

4: Quite sure. I tried to check the important points carefully. It's unlikely, though conceivable, that I missed something that should affect my ratings.

---

> ### Author Rebuttal · Authors · 2023-08-28
>
> Thank you for taking the time to review the paper draft. To your point, it’s been hard to keep up with the latest SOTA as we iterate on our experimental design, in the post-ChatGPT world we live in. We’d like to emphasize that the core contribution of this work is the largest multilingual dataset for task-oriented dialog systems with original (i.e. not translated) conversations, which also includes a high representation of disfluencies, user revisions, and code-switching. We argue that data contributions are at least as significant as modeling contributions and that the modeling and experiments part of the paper are only intended to demonstrate the utility of the data resources we carefully planned, collected, annotated, and reviewed in a systematic and ethical way as described in the paper.

---

### Meta-Review · Area_Chair_shWx · 2023-09-14

**Recommendation:** 5

**Metareview:**

The paper introduces a new multilingual dataset of task-oriented dialogues with structured contextual information (user contacts, lists).

**Pros**: All reviewers agree the dataset will be of significant value to the community, with most reviewers emphasizing the dataset's focus on phenomena which is absent from most current benchmarks; e.g., disfluencies, code-switching, and revisions. Experiments are considered "commendable" and "show real world insights into the task". The dataset is considered to be "clearly informed by [real-world] experience" and one which "enhances the realism... which is often lacking in existing datasets."

**Cons**: One reviewer raises concerns about the caliber of the model used for baselines.

---

### Decision · Program_Chairs · 2023-10-07

**Decision:**

Accept-Main

**Comment:**

The paper introduces a new multilingual dataset of task-oriented dialogues with structured contextual information (user contacts, lists).

**Pros**: All reviewers agree the dataset will be of significant value to the community, with most reviewers emphasizing the dataset's focus on phenomena which is absent from most current benchmarks; e.g., disfluencies, code-switching, and revisions. Experiments are considered "commendable" and "show real world insights into the task". The dataset is considered to be "clearly informed by [real-world] experience" and one which "enhances the realism... which is often lacking in existing datasets."

**Cons**: One reviewer raises concerns about the caliber of the model used for baselines.